# Acute effects of the different relaxation periods during passive intermittent static stretching on arterial stiffness

Yuya Higaki[1], Yosuke Yamato[1,2], Shumpei Fujie[1], Kenichiro Inoue[1,3], Mio Shimomura[1], Shosaku Kato[1], Naoki Horii[1,3], Shigehiko Ogoh[4], Motoyuki Iemitsu[1] *

1 Faculty of Sport and Health Science, Ritsumeikan University, Shiga, Japan, 2 Department of Physical Therapy, Aino University, Osaka, Japan, 3 Research Fellow of Japan Society for the Promotion of Science, Tokyo, Japan, 4 Department of Biomedical Engineering, Toyo University, Saitama, Japan

* iemitsu@fc.ritsumei.ac.jp

**Data Availability Statement:** All relevant data are within the paper and its uploaded Supporting Information files.

## Abstract

To clarify whether the relaxation period during stretching affects the degree of elevated shear rate and the degree of reduction of arterial stiffness, we examined relaxation duration to build an adequate stretching protocol. In Experiment 1, the changes in cardiac output, the shear rate in the posterior tibial artery, and blood volume in the calf muscle were measured during recovery (0–60 s) from a single bout of one-legged passive calf stretching in 12 healthy young men. In Experiment 2, the effects of different relaxation periods (5-, 10-, 20-, and 60-s) of passive one-legged intermittent calf stretching (30-s × 6 sets) on the femoral-ankle pulse wave velocity (faPWV) as an index of peripheral arterial stiffness were identified in 17 healthy young men. As a result, the stretched leg's shear rate significantly increased from 0 to 10$^{th}$ s after stretching. The muscle blood volume in the stretched leg significantly reduced during stretching, and then significantly increased during the recovery period after stretching; however, cardiac output remained unchanged during stretching and recovery. Additionally, the reduction in faPWV from the pre-stretching value in the stretched leg was significantly larger in the protocol with 10-s and 20-s relaxation periods than that in the non-stretched leg, but this did not differ in the 5-s and 60-s relaxation periods. These findings suggest that the relaxation periods of intermittent static stretching that cause a high transient increase in shear rate (via reperfusion after microvascular compression by the stretched calf muscles) are effective to reduce arterial stiffness.

## Introduction

Increased arterial stiffness is an independent risk factor for cardiovascular disease, a main causal factor of mortality worldwide [1,2]. Interestingly, habitual static stretching mitigates arterial stiffness [3–5]; thus, it has the potential to prevent cardiovascular events. Our previous studies demonstrated that even a single bout of whole-body static stretching acutely reduced systemic arterial stiffness [6]. Moreover, this decrease in arterial stiffness occurs only in the

**Funding:** This work was supported by Grants-in-Aid for Scientific Research from the Ministry of Education, Culture, Sports, Science, and Technology of Japan (KAKENHI: 19K22828 for M. Iemitsu). The funders had no role in study design, data collection and analysis, decision to publish, or preparation of the manuscript.

**Competing interests:** The authors have declared that no competing interests exist.

lower limb arteries of the stretched limb, but not in the non-stretched leg, indicating that changes in mechanical factors, rather than the hemodynamics caused by stretching induced the decrease in arterial stiffness [7]. Thus, we could expect chronic repetition of stretch stimulation to possibly lead to low basal arterial stiffness; however, the mechanism of static stretching-induced decrease in arterial stiffness remains unclear.

Continuous high shear stress (e.g., shear rate) on endothelial cells induces vasodilation with increased nitric oxide (NO) production, an endothelial-derived relaxation factor [8]. Furthermore, it has been suggested that low arterial stiffness is associated with vasodilation via endothelial functions [9]. Venturelli et al. [10] observed that acute passive quadriceps intermittent stretching, composed of 15-s relaxation after 45-s static stretching, increased blood flow in the femoral artery of the stretched leg, indicating the possibility that stretching may increase the shear rate and induce vasodilation via endothelial function. Our recent study demonstrated that acute passive one-legged calf intermittent stretching (30-s × 6 sets) significantly increased shear rate as well as blood flow in the posterior tibial artery of the stretched leg during a 10-s relaxation period after stretching, when compared to that before stretching [11]. These findings suggest that intermittent calf stretching increases shear rate during the relaxation period and consequently decreases arterial stiffness. It is noteworthy that a single bout of passive 4-min continuous calf stretching increased shear rate as well as blood flow in the popliteal artery until 15 s after stretching and returned to the baseline around 60 s after stretching [12], indicating that different relaxation periods may modify the degree of reduction in arterial stiffness as well as change in shear rate.

Increased blood flow and shear rate in the lower limb arteries of the stretched limb may be included by an increase in central circulation (cardiac output [CO]) during stretching [10,12], because an increase in central circulation is related to increased blood flow and shear rate in the peripheral arteries [13,14]. Additionally, on investigating the dynamics of blood volume in skeletal muscles, it was found that ischemic reperfusion after stretching may also be involved in the increased blood flow and shear rate after stretching [12]. Given this background, we hypothesized that changes in central circulation or different relaxation periods of intermittent stretching affect the degree of increase in shear rate, and consequently determines the degree of reduction in arterial stiffness. In the present study, to test our hypothesis, we examined time-course changes in blood flow and shear rate in the posterior tibial arteries of passive calf-stretched and non-stretched legs, especially during 30-s stretching and 60-s relaxation periods (Experiment 1, Exp. 1). In addition, we measured CO and muscle blood volume to examine the effect of central and peripheral circulation on the stretching-induced increase in shear rate. Moreover, to identify the effects of the different relaxation durations of intermittent stretching on arterial stiffness, we measured pulse wave velocity (PWV) as an index of arterial stiffness in the stretched leg in 5-, 10-, 20-, and 60-s relaxation periods of acute passive one-legged intermittent calf stretching protocol in healthy young men (Experiment 2, Exp. 2). These investigations may provide important information to build an adequate stretching protocol to efficiently reduce arterial stiffness.

## Methods

### Experiment 1

**Subjects.** Twelve healthy young men (age, 22.8 ± 0.3 years; height, 174.1 ± 1.7 cm; weight, 65.1 ± 1.6 kg; body mass index, 21.6 ± 0.5 kg/m$^2$) participated in this study. Participants were recruited by flyers and poster placed on the local community. None of the participants habitually exercised (e.g., aerobic and/or resistance training), had cardiovascular disease, were smokers, or used prescription or over-the-counter medications. All participants provided written,

## A: Experiment 1

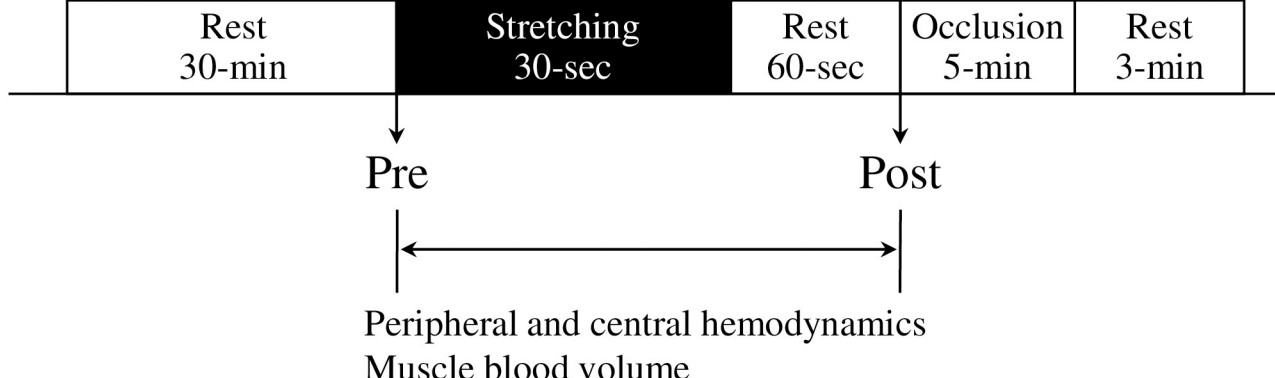

## B: Experiment 2

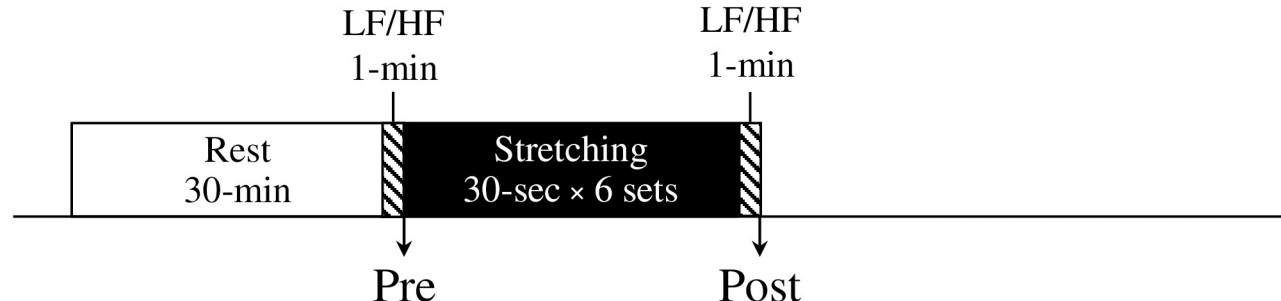

**Fig 1.** Design of experiments 1 (A) and 2 (B). In Experiment 1, blood pressure is measured before stretching (Pre) and after stretching (Post). Peripheral (artery diameter, mean blood velocity, blood flow, shear rate) and central (heart rate, stroke volume, cardiac output) hemodynamics and muscle blood volume (total hemoglobin) are measured during the stretching session. In Experiment 2, brachial-ankle pulse wave velocity (baPWV), femoral-ankle PWV (faPWV), heart rate, blood pressures are measured at Pre and Post. low frequency (LF)/high frequency (HF) is measured for 1-min immediately before and after stretching.

informed consent before their participation in this study, which was approved by the Ethics Committee of Ritsumeikan University and was conducted in accordance with the Declaration of Helsinki.

**Experimental design.** All participants refrained from caffeine, alcohol, and strenuous physical activity for 24 h, which was followed by a 6-hour fast (water could be taken during the fast). The height and body weight (InBody770; Inbody Co., Ltd., Seoul, Korea) were then measured (Fig 1A). The diameter and mean blood velocity in the posterior tibial artery, the heart rate (HR), stroke volume (SV), blood pressures, and total hemoglobin (Hbtot) as an index of blood volume in the stretched skeletal muscle were measured in the supine position after at least 30 min of rest. Posterior tibial artery diameter, mean blood velocity, HR, SV, and Hbtot were measured in the supine position during stretching and till 60 s after stretching. Additionally, blood pressures were measured 60 s after stretching. To normalize the concentration of oxygenated hemoglobin and myoglobin ($O_2Hb$) and the concentration of deoxygenated hemoglobin and myoglobin (HHb), a pneumatic cuff placed proximal to a near-infrared

spectroscopy (NIRS) sensor was inflated for 5 min, followed by rest for 3 min after release. The posterior tibial artery diameter, mean blood velocity, HR, SV, and Hbtot were measured continuously and calculated every 5 s for each point. The room temperature was maintained at $24 \pm 1°C$ during the experiment.

**Stretching protocol.** A single bout of passive one-legged calf static stretching was passively held for 30 s at the end range (point of minimal discomfort), followed by a 60-s rest period. In each participant, the stretched and non-stretched legs were randomized. The investigator passively moved the ankle joint of the stretched leg slowly (for at least 5 s) through the full range of motion. In contrast, the non-stretched leg was not moved.

**Measurements of peripheral hemodynamics, HR, SV, and blood pressures.** Measurements of the posterior tibial artery diameter and mean blood velocity were performed in the passively stretched and non-stretched legs, behind the malleolus medialis of the tibia, using the Vivid S6 and q ultrasound system (GE Healthcare, Chicago, USA). Each Vivid S6 and q ultrasound system was equipped with a linear array transducer operating at an imaging frequency of 10 MHz. Posterior tibial artery diameter was determined at an angle perpendicular to the central axis of the scanned area with B-mode imaging. Blood velocity was measured using the PW mode of the same probe at a frequency of 4.2 MHz. All blood velocity measurements were obtained with the probe appropriately positioned to maintain an insonation angle of $\leq 60°$. The sample volume was maximized according to the vessel size and centered within the vessel based on real-time ultrasound visualization. The arterial diameter was measured, and mean blood velocity values were automatically calculated using the available software (Vivid S6 and q). The diameter and mean blood velocity were used to calculate the posterior tibial artery blood flow and shear rate. Blood flow was calculated as: mean blood velocity $\times \pi$ (vessel diameter/2)$^2 \times 60$ [11,12,15], where blood flow was measured in milliliters per minute (mL/min). Shear rate (per second), a useful estimator of shear stress that does not account for blood viscosity, was defined as: [4 $\times$ mean blood velocity]/vessel diameter [11,12,15]. The aortic diameter and blood velocity were measured using the LOGIQ S7 ultrasound system (GE Healthcare, Chicago, USA). HR was automatically calculated using the LOGIQ S7 software before, during, and after the stretching session. The LOGIQ S7 ultrasound system was equipped with a sector scanner transducer operating at an imaging frequency of 2.5 MHz. The aortic diameter was measured at end-diastole during three consecutive cardiac cycles by the leading edge-to-leading edge method, and the mean diameter was determined according to a previous study [16]. The time-velocity integral was derived by digitizing the blood velocity curve, as outlined in the second cardiac cycle, according to a previous study [16]. The SV was calculated as: time-velocity integral $\times \pi$ (aortic diameter/2)$^2$; and CO was calculated as: SV $\times$ HR at each point [16]. Systolic and diastolic blood pressures were measured using an automatic blood pressure measuring device (HEM-762; OMRON, Kyoto, Japan) at the right brachial artery immediately before and after the stretching session.

**Measurements of blood volume of gastrocnemius medialis muscle.** Microvascular $O_2Hb$ and $HHb$ were measured using a frequency-domain phase modulation near-infrared spectrophotometer (NIRO-200NX; Hamamatsu Photonics K.K., Shizuoka, Japan) at the medial head of the gastrocnemius muscle in the stretched leg. The NIRS probe was placed at one-third the distance between the popliteal crease and the medial malleolus and held in place by elastic tape. NIRS provides continuous, non-invasive monitoring of the relative concentration changes of Hbtot ($O_2Hb$ + $HHb$) as an index of muscle blood volume before, during, and after stretching. Hbtot was assessed at baseline after a 30-min rest, followed by a single bout of stretching of the gastrocnemius muscle. Simultaneously, Hbtot was assessed before, during, and after stretching every 5 s. To normalize the $O_2Hb$ and $HHb$ for each subject, a pneumatic cuff (AG101 & E20; D.E. Hokanson Inc., Bellevue, USA) was placed proximal to the NIRS

sensor and inflated to 250 mmHg for 5-min, or until $O_2Hb$ and HHb reached a plateau, and rested for 3-min after release [12]. The highest value of $O_2Hb$ after release was defined as 100%, and the minimum value during ischemia was defined as 0%. The lowest value of HHb after release was defined as 0%, and the maximum value during ischemia was defined as 100%. NIRS measurements of $O_2Hb$ and HHb before, during, and after the stretching session were then calculated as relative percent concentration changes of Hbtot.

**Statistical analysis.**　Values are expressed as mean ± standard error (SE). Two-way (leg × time) repeated-measures analysis of variance (ANOVA) was performed on the posterior tibial artery diameter, mean blood velocity, blood flow, and shear rate. One-way repeated-measures ANOVA was performed on SV, HR, CO, and Hbtot. If measurements at each time point were significantly different, Fisher's post hoc test was applied. The t-test was used to compare blood pressures before and after stretching. Statistical significance was set at $P < 0.05$. All statistical analyses were performed using StatView 5.0 (SAS Institute, Tokyo, Japan). Using an alpha error of 0.05, a power (1-β) of 0.80, the effect size of 0.25 [17], the number of groups of 2, and the number of measurements of 20, a sample size resulted in five subjects (G*Power 3.1).

## Experiment 2

**Subjects.**　Seventeen healthy young men (age, 23.1 ± 0.4 years; height, 173.5 ± 1.4 cm; weight, 67.7 ± 2.3 kg; body mass index, 22.5 ± 0.8 kg/m$^2$) participated in this study. Participants were recruited by flyers and poster placed on the local community. None of the participants habitually exercised (e.g., aerobic and/or resistance training), had cardiovascular disease, were smokers, or used prescription or over-the-counter medications. All participants provided written, informed consent before their participation in the study, which was approved by the Ethics Committee of Ritsumeikan University and was conducted in accordance with the Declaration of Helsinki.

**Experimental design.**　All participants refrained from caffeine, alcohol, and strenuous physical activity for 24 h, which was followed by a 6-hour fast (water could be taken during the fast). The height and body weight (InBody770; Inbody Co., Ltd.) were measured, and a sit-and-reach test (T.K.K.5112; Takei Scientific Instruments Co., Ltd, Tokyo, Japan) was performed (Fig 1B). PWV, blood pressures, HR, and low frequency (LF)/high frequency (HF) as an index of sympathetic nerve activity were measured in the supine position after at least 30-min of rest (Pre) and 1-min immediately after stretching (Post). The room temperature was maintained at 24 ± 1˚C during the experiment.

**Stretching protocol.**　The participants experienced a single session of passive one-legged calf static stretching, as previously described [11]. In this study, stretching was passively held for 30 s at the end range (point of minimal discomfort), followed by 5-, 10-, 20-, or 60-s relaxation periods for six repetitions. In each participant, the stretched and non-stretched legs, and the order in which relaxation period conditions were performed, were randomized. The investigator passively moved the ankle joint of the stretched leg slowly (at least 5 s) through the full range of motion. In contrast, the non-stretched leg was not moved.

**Measurements of PWV, blood pressures, and HR.**　Bilateral brachial and ankle blood pressures, PWV, and HR were concurrently measured using a vascular testing device (form PWV/ABI; Omron Colin, Kyoto, Japan). Bilateral brachial and posterior tibial arterial pressure waveforms were observed using oscillometric pressure sensors wrapped on both arms and ankles. Femoral arterial pressure waveforms were acquired by placing an applanation tonometry sensor on the left femoral artery, as previously described [6,7,18]. Electrocardiogram electrodes were fixed on both wrists, and a microphone was positioned on the left edge of the sternum to measure the HR. PWV was calculated as the distance between two arterial measuring sites divided by the transit time. The transit time was determined from the time delay

between the proximal and distal "foot" waveforms. The foot of the wave was identified as the point of commencement of the sharp systolic upstroke, which was observed automatically. The following PWVs were assessed: brachial-ankle PWV (baPWV) as an index of systemic arterial stiffness, and femoral-ankle PWV (faPWV) as an index of peripheral arterial stiffness.

**Heart rate variability assessment.** HR variability (HRV) was continuously measured using an R-R monitor (Polar V800, Poral Electro, Kempele, Finland) for 1 min immediately before and after stretching. The sensor-measured HRV was placed at the front of the chest using an exclusive band. The acquired data were analyzed using Kubios HRV Standard version.3.3 (Kubios, Kuopio, Finland). The area underneath the spectral bands within the range of 0.04–0.15 Hz was defined as LF, and the area underneath the spectral band within the range of 0.15–0.40 Hz was defined as HF. LF/HF, as an index of parasympathetic activity, was calculated before and immediately after stretching.

**Statistical analysis.** Values are expressed as mean ± SE. A three-way (leg × time × condition) repeated-measures ANOVA was performed on the baPWV, faPWV, and blood pressures. A two-way (time × condition) repeated-measures ANOVA was performed on the HR, LF/HF, sit-and-reach test. A two-way (leg × condition) repeated-measures ANOVA was performed on the amount of change in faPWV [(faPWV after stretching)–(faPWV before stretching)] of the stretched and non-stretched legs. If measurements of each condition were significantly different, Fisher's post-hoc test was applied. Statistical significance was set at $P < 0.05$. All statistical analyses were performed using StatView 5.0 (SAS Institute). Using an alpha error of 0.05, a power (1-β) of 0.80, the effect size of 0.25 [17], the number of groups of 2, and the number of measurements of 4, a sample size resulted in eleven subjects (G*Power 3.1). Cohen's d was calculated to evaluate the effect size between the 5-s relaxation period and other periods in the amount of change in faPWV in the stretched leg (G*Power 3.1).

## Results

### Experiment 1

**Response to blood pressures before and after stretching.** Brachial systolic blood pressure (SBP) and diastolic blood pressure (DBP) did not differ between immediately before and after the stretching session (SBP: pre, 112.8 ± 1.9 mmHg; post, 110.8 ± 1.5 mmHg; DBP: pre, 69.4 ± 2.0 mmHg, post, 67.3 ± 1.2 mmHg).

**Response to peripheral hemodynamics before, during, and after stretching.** The posterior tibial artery diameter in the stretched leg was significantly reduced during 5 s to 30 s of stretching (5 s, $P = 0.010$; 10 s, $P = 0.003$; 15 s, $P = 0.006$; 20 s, $P = 0.004$; 25 s, $P = 0.003$; 30 s, $P = 0.004$), and significantly increased 40 s after stretching compared to that during pre ($P = 0.039$) (Fig 2A). It also significantly reduced during 5 s to 30 s of stretching (5 s, $P = 0.003$; 10 s, $P < 0.001$; 15 s, $P = 0.002$; 20 s, $P = 0.009$; 25 s, $P = 0.003$; 30 s, $P = 0.001$), and significantly increased 20 s to 55 s after stretching (20 s, $P = 0.022$; 25 s, $P = 0.012$; 30 s, $P = 0.012$; 35 s, $P = 0.002$; 40 s, $P < 0.001$; 45 s, $P = 0.012$; 50 s, $P = 0.008$; 55 s, $P = 0.018$) compared to that in the non-stretched leg (Fig 2A; interaction, $P < 0.0001$). The posterior tibial artery mean blood velocity in the stretched leg was significantly increased 0 to 10 s after stretching compared to that during pre ($P < 0.0001$) (Fig 2B). It also significantly reduced at 5-s and 20-s during stretching (5-s, $P = 0.042$; 20-s, $P = 0.042$), and significantly increased 0 to 10 s after stretching (0 s, $P = 0.014$; 5 s, $P = 0.009$; 10 s $P = 0.019$) compared to that in the non-stretched leg (Fig 2B; interaction, $P < 0.0001$). The posterior tibial artery blood flow in the stretched leg was significantly increased at 0 to 10-s and 30-s after stretching compared to that during pre (0 s, $P = 0.034$; 5 s, $P = 0.002$; 10 s, $P = 0.010$; 30 s, $P = 0.047$) (Fig 2C). It also significantly reduced during 5 s to 30 s of stretching (5 s, $P = 0.018$; 10 s, $P = 0.013$; 15 s, $P = 0.012$; 20 s, $P = 0.001$; 25

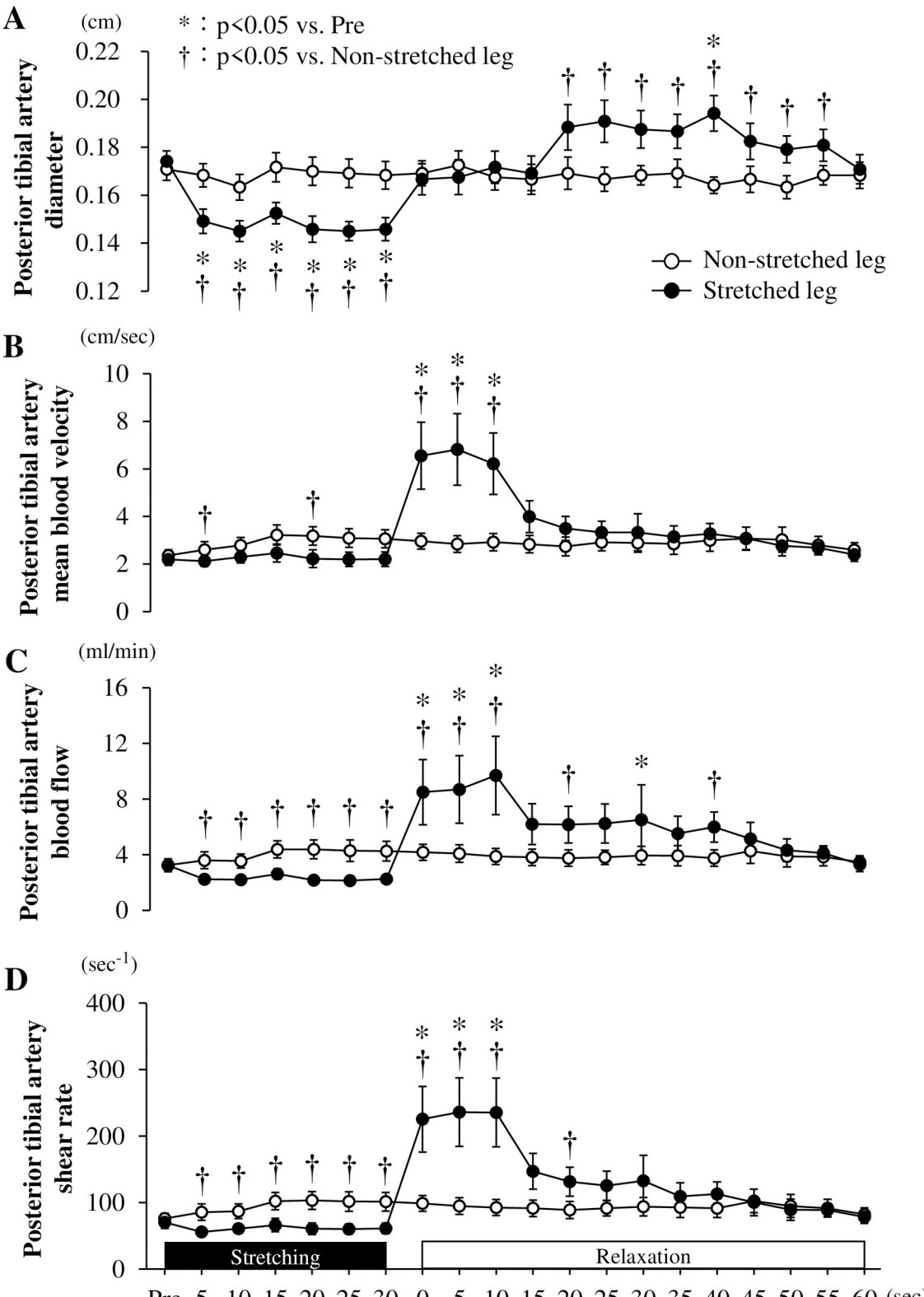

**Fig 2.** Artery diameter (A), mean blood velocity (B), blood flow (C), and shear rate (D) of the posterior tibial artery before, during, and after passive one-legged calf stretching. Pre, before stretching; * $P < 0.05$ vs. Pre, † $P < 0.05$ vs. Non-stretched leg. Data are presented as mean ± SE.

s, $P = 0.008$; 30 s, $P = 0.003$), and significantly increased at 0 to 10-s, 20-s, and 40-s after stretching (0 s, $P = 0.049$; 5 s, $P = 0.035$; 10 s, $P = 0.047$; 20 s, $P = 0.028$; 40 s, $P = 0.016$) compared to that in the non-stretched leg (Fig 2C; interaction, $P < 0.01$). The posterior tibial artery shear rate in the stretched leg was significantly increased 0 to 10 s after stretching compared to that during pre (0 s, $P = 0.001$; 5 s, $P < 0.001$; 10 s, $P < 0.001$) (Fig 2D). It also significantly reduced at 5-s to 30-s during stretching (5 s, $P = 0.017$; 10 s, $P = 0.030$; 15 s, $P = 0.032$; 20 s, $P = 0.003$; 25 s, $P = 0.013$; 30 s, $P = 0.008$), and significantly increased at 0 to 10 s, and 20 s after stretching (0 s, $P = 0.030$; 5 s, $P = 0.019$; 10 s, $P = 0.030$; 20 s, $P = 0.030$) compared to that in the non-stretched leg (Fig 2D; interaction, $P < 0.0001$).

**Response to central hemodynamics before, during, and after stretching.** SV, HR, and CO did not change during and after stretching (Fig 3A–3C).

**Response to Hbtot before, during, and after stretching.** Hbtot was significantly reduced at 5 s to 30 s during stretching (5 s, $P = 0.049$; 10 s, $P = 0.031$; 15 s, $P = 0.029$; 20 s, $P = 0.020$; 25 s, $P = 0.020$; 30 s, $P = 0.028$) and significantly increased 5-s to 60-s after stretching (5 s, $P = 0.039$; 10 s, $P = 0.031$; 15 s, $P = 0.042$; 20 s, $P = 0.036$; 25 s, $P = 0.039$; 30 s, $P = 0.037$; 35 s, $P = 0.048$; 40 s, $P = 0.039$; 45 s, $P = 0.032$; 50 s, $P = 0.029$; 55 s, $P = 0.026$; 60 s $P = 0.023$) compared to pre-stretching (Fig 4).

### Experiment 2

Next, we performed Exp. 1 to investigate the effect of arterial stiffness on different relaxation periods after stretching.

**Response to blood pressures, baPWV, faPWV, HR, and LF/HF before and after stretching.** Blood pressures, baPWV, faPWV, HR, and LF/HF did not differ between before and after stretching, under all conditions of relaxation (Tables 1 and 2). The amount of change in faPWV in the stretched leg was significantly reduced in the 10-s and 20-s relaxation periods compared to that in the non-stretched leg (Fig 5; 10-s, $P = 0.011$; 20-s, $P = 0.002$), but did not differ between stretched and non-stretched legs in the 10- and 20-s relaxation periods. Additionally, the amount of change in faPWV in the stretched leg was significantly reduced in the 10-s, 20-s, and 60-s relaxation periods compared to that in the 5-s relaxation period (10-s: $P = 0.022$, $d = 0.95$; 20-s: $P < 0.001$, $d = 1.54$; 60-s: $P = 0.007$, $d = 0.86$) (Fig 5).

**Response to sit-and-reach test before and after the stretching.** Sit-and-reach test under all conditions did not differ between before and after stretching (5-s relaxation period: Pre, $40.3 \pm 1.5$ cm; Post, $38.8 \pm 1.7$ cm; 10-s relaxation period: Pre, $39.0 \pm 1.7$ cm, Post, $38.6 \pm 1.8$ cm; 20-s relaxation period: Pre, $41.4 \pm 1.5$ cm, Post, $40.9 \pm 1.5$ cm; and 60-s relaxation period: Pre, $40.5 \pm 1.8$ cm, Post, $40.1 \pm 1.8$ cm).

## Discussion

In Exp. 1 of the present study, we revealed that the blood flow and shear rate in the posterior tibial artery increased from the baseline immediately (10 s) after 30 s of stretching. However, the CO and blood pressures were not affected by stretching. These findings indicate that mild ischemia-reperfusion was caused mechanically by the stretching calf muscles rather than by central hemodynamics, and consequently, it may have induced the increases in blood flow and shear rate in the posterior tibial artery of the stretched leg. Additionally, in Exp. 2, the PWV after acute passive one-legged intermittent calf stretching (30 s × 6 sets) was decreased in the protocols using relaxation periods of 10 s to 20 s but not in those using short (5 s) or long relaxation periods (60 s). These findings suggest that the degree of decrease in arterial stiffness in the intermittent stretching protocol is likely due to a degree of increased shear rate, which is determined by the relaxation duration of the intermittent stretching protocol.

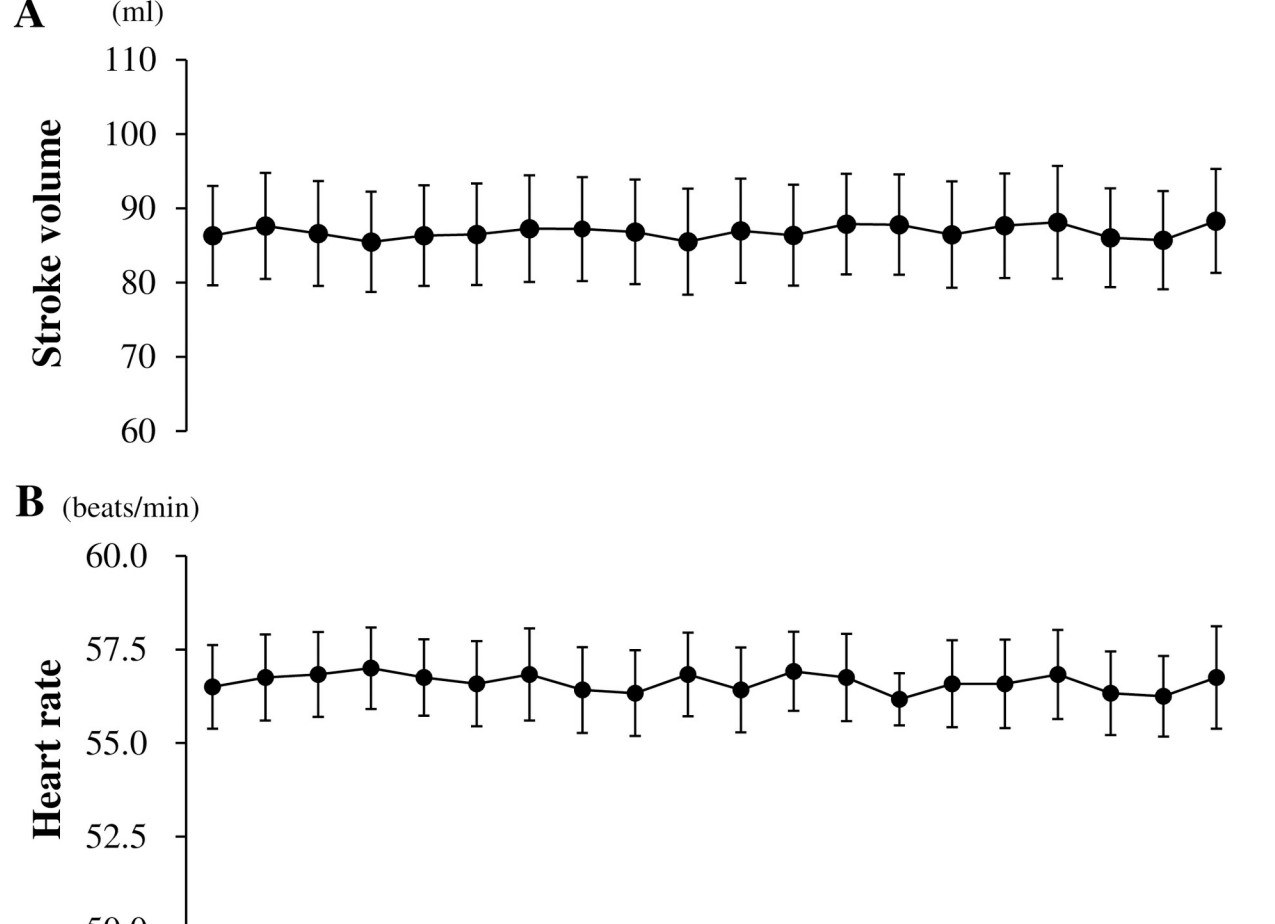

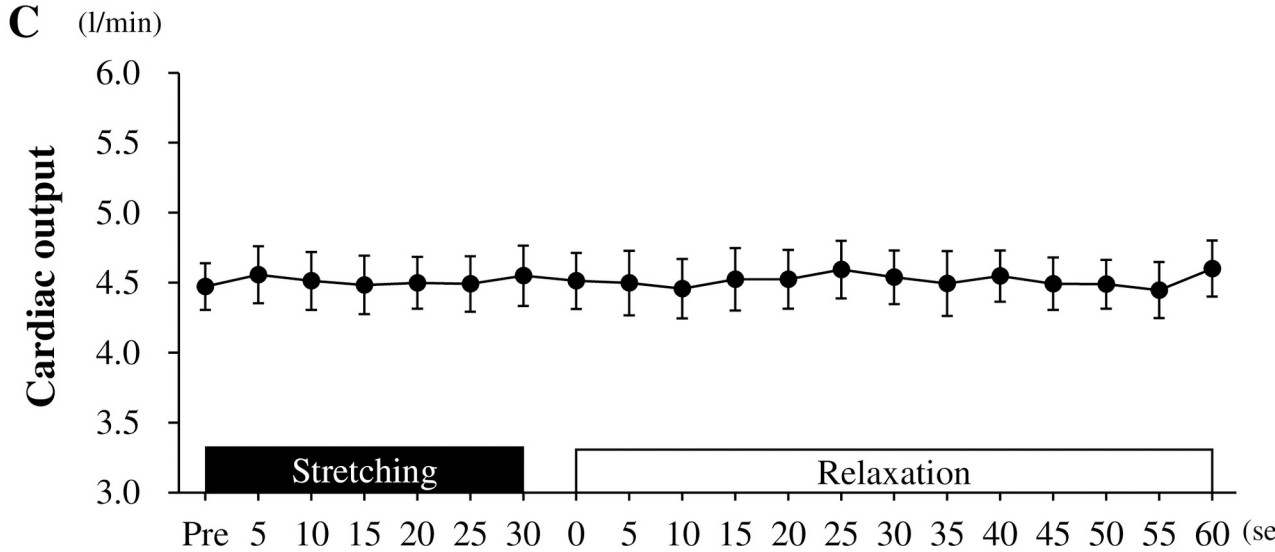

**Fig 3.** Stroke volume (A), heart rate (B), and cardiac output (C) before, during, and after passive one-legged calf stretching. Pre, before stretching; Data are presented as mean ± SE.

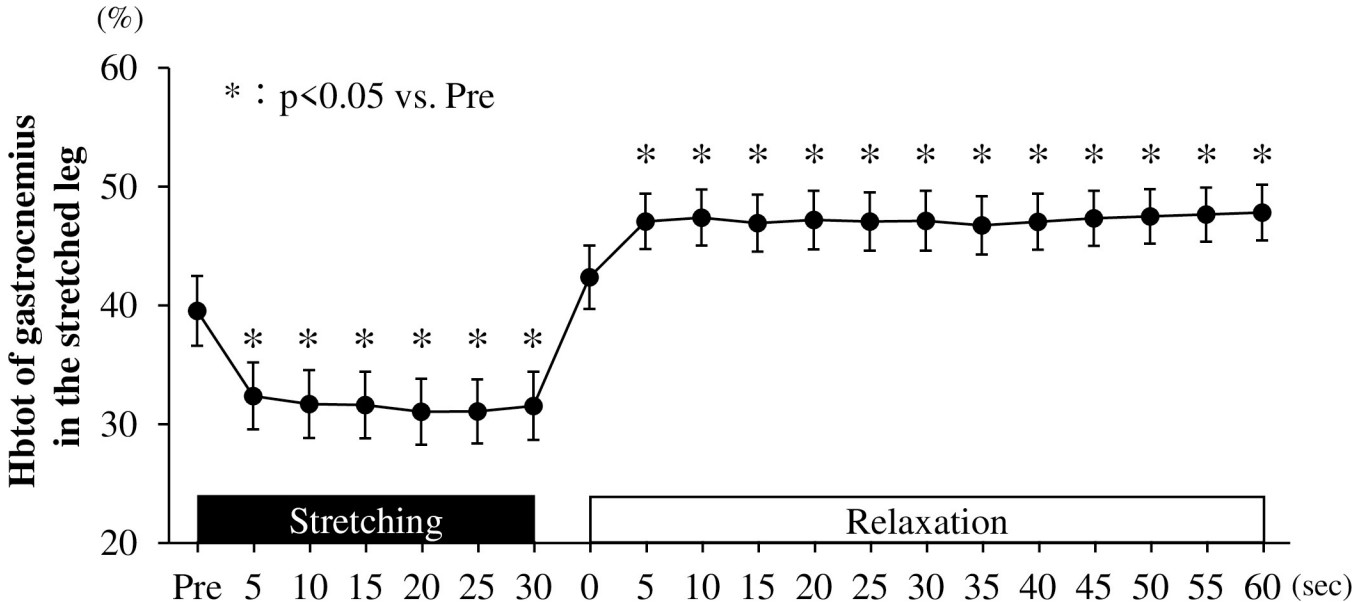

**Fig 4. Total hemoglobin (Hbtot) of the gastrocnemius muscle in the stretched leg before, during, and after passive one-legged calf stretching.** Pre, before stretching; * $P < 0.05$ vs. Pre. Data are presented as mean ± SE.

**Table 1. Blood pressures, baPWV, and faPWV before (Pre) and after (Post) passive one-legged calf stretching.**

|  | 5-sec | | | | | | 10-sec | | | | | | 20-sec | | | | | | 60-sec | | | | | |
|---|---|---|---|---|---|---|---|---|---|---|---|---|---|---|---|---|---|---|---|---|---|---|---|---|
|  | Pre | | | Post | | | Pre | | | Post | | | Pre | | | Post | | | Pre | | | Post | | |
| bSBP (mmHg) | | | | | | | | | | | | | | | | | | | | | | | | |
| Con | 110 | ± | 3 | 111 | ± | 2 | 109 | ± | 2 | 109 | ± | 2 | 110 | ± | 2 | 109 | ± | 1 | 109 | ± | 2 | 109 | ± | 2 |
| Stretch | 108 | ± | 2 | 109 | ± | 2 | 111 | ± | 2 | 110 | ± | 2 | 109 | ± | 2 | 108 | ± | 1 | 109 | ± | 2 | 108 | ± | 2 |
| bDBP (mmHg) | | | | | | | | | | | | | | | | | | | | | | | | |
| Con | 62 | ± | 3 | 62 | ± | 2 | 61 | ± | 1 | 62 | ± | 2 | 62 | ± | 1 | 61 | ± | 2 | 62 | ± | 2 | 63 | ± | 3 |
| Stretch | 62 | ± | 2 | 61 | ± | 2 | 62 | ± | 1 | 62 | ± | 2 | 61 | ± | 2 | 60 | ± | 1 | 63 | ± | 2 | 62 | ± | 2 |
| aSBP (mmHg) | | | | | | | | | | | | | | | | | | | | | | | | |
| Con | 126 | ± | 3 | 127 | ± | 3 | 127 | ± | 2 | 127 | ± | 3 | 125 | ± | 3 | 126 | ± | 3 | 126 | ± | 2 | 127 | ± | 3 |
| Stretch | 125 | ± | 3 | 127 | ± | 3 | 124 | ± | 3 | 126 | ± | 3 | 127 | ± | 2 | 126 | ± | 3 | 121 | ± | 2 | 124 | ± | 2 |
| aDBP (mmHg) | | | | | | | | | | | | | | | | | | | | | | | | |
| Con | 65 | ± | 2 | 65 | ± | 2 | 67 | ± | 2 | 66 | ± | 2 | 63 | ± | 2 | 64 | ± | 2 | 67 | ± | 2 | 68 | ± | 2 |
| Stretch | 66 | ± | 3 | 66 | ± | 2 | 65 | ± | 2 | 64 | ± | 2 | 64 | ± | 2 | 65 | ± | 2 | 65 | ± | 2 | 66 | ± | 2 |
| baPWV (cm/sec) | | | | | | | | | | | | | | | | | | | | | | | | |
| Con | 1035 | ± | 27 | 1036 | ± | 28 | 1035 | ± | 27 | 1020 | ± | 29 | 1028 | ± | 24 | 1011 | ± | 27 | 1030 | ± | 22 | 999 | ± | 23 |
| Stretch | 1034 | ± | 31 | 1032 | ± | 29 | 1037 | ± | 29 | 1000 | ± | 28 | 1045 | ± | 28 | 1001 | ± | 29 | 1052 | ± | 21 | 1011 | ± | 26 |
| faPWV (cm/sec) | | | | | | | | | | | | | | | | | | | | | | | | |
| Con | 854 | ± | 23 | 862 | ± | 24 | 853 | ± | 27 | 849 | ± | 29 | 862 | ± | 26 | 851 | ± | 27 | 846 | ± | 26 | 822 | ± | 24 |
| Stretch | 855 | ± | 29 | 857 | ± | 26 | 857 | ± | 29 | 829 | ± | 28 | 881 | ± | 32 | 838 | ± | 29 | 868 | ± | 23 | 835 | ± | 27 |

5-sec, condition of relaxation period 5-sec: 10-sec, condition of relaxation period 10-sec: 20-sec, condition of relaxation period 20-sec: 60-sec, condition of relaxation period 60-sec: Pre, before stretching: Post, 1-min immediately after stretching: bSBP, branchial systolic blood pressure: bDBP, brachial diastolic blood pressure: aSBP, ankle systolic blood pressure: aDBP, ankle diastolic blood pressure: baPWV, brachial-ankle pulse wave velocity: faPWV, femoral-ankle pulse wave velocity: Con, non-stretch side: Stretch, stretch side; Data are presented as mean±SE.

**Table 2. HR and LF/HF before (Pre) and after (Post) passive one-legged calf stretching.**

| | 5-sec | | | | | | 10-sec | | | | | | 20-sec | | | | | | 60-sec | | | | | |
| --- | --- | --- | --- | --- | --- | --- | --- | --- | --- | --- | --- | --- | --- | --- | --- | --- | --- | --- | --- | --- | --- | --- | --- | --- |
| | Pre | | | Post | | | Pre | | | Post | | | Pre | | | Post | | | Pre | | | Post | | |
| HR (beats/min) | 52.9 | ± | 1.5 | 52.5 | ± | 1.6 | 51.3 | ± | 1.4 | 51.4 | ± | 1.7 | 52.6 | ± | 1.6 | 51.9 | ± | 1.4 | 52.7 | ± | 1.5 | 51.8 | ± | 1.5 |
| LF/HF | 2.0 | ± | 0.3 | 1.5 | ± | 0.3 | 1.1 | ± | 0.2 | 2.2 | ± | 0.7 | 1.8 | ± | 0.5 | 3.0 | ± | 0.9 | 2.1 | ± | 0.6 | 3.1 | ± | 1.1 |

5-sec, condition of relaxation period 5-sec: 10-sec, condition of relaxation period 10-sec: 20-sec, condition of relaxation period 20-sec: 60-sec, condition of relaxation period 60-sec: Pre, before stretching: Post, 1-min immediately after stretching: HR, heart rate: LF, low frequency: HF, high frequency; Data are presented as mean±SE.

In this study, the blood flow and shear rate in the posterior tibial artery increased from the 0 to 10 s after stretching. Furthermore, the acute passive calf intermittent static stretching protocol (30 s × 6 sets) using 10- and 20-s relaxation periods induced a reduction in arterial stiffness in the stretched leg, but this was not seen in the protocol using 5-s and 60-s relaxation periods. These findings indicate that intermittent stretching-induced arterial stiffness and arterial shear rate are likely affected by the relaxation period. Thus, the intermittent stretching protocol should use adequate relaxation time to achieve a greater shear rate and to thus obtain a greater reduction in arterial stiffness.

Continuous high shear stress (rate) on endothelial cells acutely causes the release of NO via activation of endothelial NO synthase [8,19–21], leading to reduction of arterial stiffness with vasodilation [22]. Flow-mediated dilation (FMD), as an index of endothelial function, was identified during the rapid increase in blood flow caused by reperfusion after forearm ischemia, and it is well established that during this measurement, the shear rate acutely increases

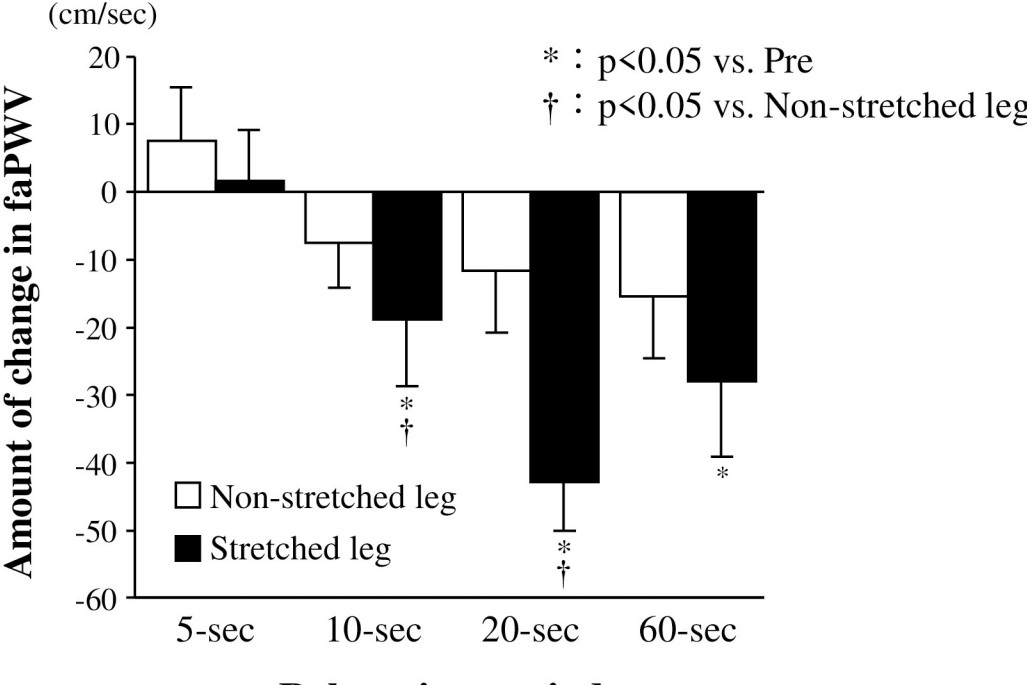

**Fig 5. Amount of change in faPWV before and after passive one-legged calf stretching.** 5-s, condition of 5-s relaxation; 10-s, condition of 10-s relaxation; 20-s, condition of 20-s relaxation; 60-s, condition of 60-s relaxation; faPWV, femoral-ankle pulse wave velocity; * $P < 0.05$ vs. 5-s, † $P < 0.05$ vs. Non-stretched leg. Data are presented as mean ± SE.

immediately after reperfusion, and afterwards, the peak increase in arterial diameter for the FMD value occurs around 20 s to 30 s after reperfusion [9]. In other words, there is a time lag between the increase in shear rate and peak vasodilation (~ 40 s) [23]. Similarly, in the present study, a reduction in arterial stiffness occurred after an increase in the shear rate, indicating that the relaxation duration that achieved a higher shear rate likely caused the higher reduction in arterial stiffness. Therefore, the stretching-induced reduction in arterial stiffness may be related to vasodilation via increased shear rate. A recent study has shown that chronic one-legged stretching increased popliteal artery FMD in the stretched leg followed by an increase in that in the non-stretched leg [5]. The acute stretching in this study affected only the stretching leg. Therefore, the local effect of the acute stretching on the whole body may occur later.

An increase in central circulation is related to an increase in vasodilation and a reduction in arterial stiffness in the peripheral arteries [13,14]. For example, it has been reported that increased central hemodynamics lead to peripheral vasodilation via the arterial and cardiopulmonary baroreflex, and this autonomic function-induced peripheral vasodilation results in a reduction of peripheral arterial stiffness [24]. Indeed, acute intermittent stretching-induced increase in central hemodynamics have been shown to increase blood flow in the stretch legs during the relaxation duration [10]. However, in this study, the blood flow in the posterior tibial arteries increased for 10 s after the 30-s passive calf intermittent static stretching, but significant changes in HR and CO were not observed during both the stretching and relaxation periods. This result is consistent with our previous study [11], which reported no change in HR during the stretching and relaxation periods in the same 30-s passive calf intermittent static stretching. Moreover, this study showed that parasympathetic activity, estimated by LF/HF, did not change before and after the acute passive calf intermittent static stretching (30 s × 6 sets) which used different relaxation periods, and it concomitantly did not alter HR, SBP, and DBP. These findings suggest that the increase in peripheral blood flow and shear rate after passive calf intermittent static stretching may be caused by a local mechanism, independent of central hemodynamics and autonomic function. Therefore, the difference in peripheral arterial stiffness responses to the different relaxation periods after stretching is independent of central hemodynamics and is related autonomic regulation.

Kruse et al. [25] reported that the blood volume of the gastrocnemius muscle decreased during stretching of the plantar flexors. Similarly, in this study, the blood volume in the gastrocnemius muscle of the calf-stretched leg decreased during stretching and increased after stretching. In addition, in previous studies, blood flow in the artery distal to the site of occlusion increased with the increase in blood flow in the vascular bed after the release of arterial occlusion [26]. Based on these findings, we speculate that static stretching may compress the capillaries of the stretched muscle and cause congestion, and release this congestion after stretching, resulting in increased blood flow to the vascular bed of the muscle. Indeed, in this study, the diameter of the posterior tibial artery was reduced during stretching, and subsequently, blood flow in the posterior tibial artery increased after stretching. Thus, the stretch-induced mild occlusion in the gastrocnemius muscle may contribute to increased blood flow in the posterior tibial artery and increase blood flow to the vascular bed in the gastrocnemius muscle after stretching. However, further investigations are needed to address this important speculation to identify the mechanism of stretch-induced reduction in arterial stiffness. Additionally, in this study, the value of the sit-and-reach test did not differ between before and after the stretching, therefore, it was not an accurate test to assess the effect of flexibility by calf static stretching. Therefore, further study needs to examine measuring the range of motion of the ankle joint after calf static stretching.

In conclusion, the degree of decrease in arterial stiffness in the intermittent stretching protocol is associated with the relaxation duration rather than central hemodynamics because the

relaxation duration modifies the shear rate. Thus, the relaxation duration should be considered while building an adequate stretching protocol to efficiently reduce arterial stiffness.

## Supporting information

**S1 Dataset.**
(XLSX)

**S2 Dataset.**
(XLSX)

## Acknowledgments

We would like to thank the participants in this study.

## Author Contributions

**Conceptualization:** Yuya Higaki, Yosuke Yamato, Motoyuki Iemitsu.

**Formal analysis:** Yuya Higaki, Yosuke Yamato, Shumpei Fujie, Naoki Horii, Motoyuki Iemitsu.

**Funding acquisition:** Motoyuki Iemitsu.

**Investigation:** Yuya Higaki, Shumpei Fujie, Kenichiro Inoue, Mio Shimomura, Shosaku Kato, Motoyuki Iemitsu.

**Project administration:** Motoyuki Iemitsu.

**Supervision:** Motoyuki Iemitsu.

**Writing – original draft:** Yuya Higaki, Yosuke Yamato, Shumpei Fujie, Kenichiro Inoue, Mio Shimomura, Naoki Horii, Shigehiko Ogoh, Motoyuki Iemitsu.

**Writing – review & editing:** Yuya Higaki, Yosuke Yamato, Shumpei Fujie, Kenichiro Inoue, Mio Shimomura, Shosaku Kato, Naoki Horii, Shigehiko Ogoh, Motoyuki Iemitsu.

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
