## [Decision Letter · Decision Letter 0]

31 Aug 2021

PONE-D-21-23457

Acute effects of the different relaxation periods during passive intermittent static stretching on arterial stiffness

PLOS ONE

Dear Dr. Iemitsu,

Thank you for submitting your manuscript to PLOS ONE. After careful consideration, we feel that it has merit but does not fully meet PLOS ONE’s publication criteria as it currently stands. Therefore, we invite you to submit a revised version of the manuscript that addresses the points raised during the review process.

All issues raised by reviewers are required.

We look forward to receiving your revised manuscript.

Kind regards,

Vincenzo Lionetti, M.D., PhD

Academic Editor

PLOS ONE

Journal Requirements:

2. In the Methods section of the manuscript, please provide additional information regarding how participants were recruited for both of the experiments within the study.

Reviewers' comments:

Reviewer's Responses to Questions

**Comments to the Author**

1. Is the manuscript technically sound, and do the data support the conclusions?

Reviewer #1: Yes

Reviewer #2: Yes

2. Has the statistical analysis been performed appropriately and rigorously? 

Reviewer #1: No

Reviewer #2: Yes

3. Have the authors made all data underlying the findings in their manuscript fully available?

Reviewer #1: No

Reviewer #2: Yes

4. Is the manuscript presented in an intelligible fashion and written in standard English?

Reviewer #1: Yes

Reviewer #2: Yes

5. Review Comments to the Author

Reviewer #1: The manuscript is focused on a interesting topic and could have some important practical implication.

The Authors tired to find the best relaxation period to optimise the vascular responsiveness (including measures of shear rare and arterial stiffness) of the tibial posterior artery after an unilateral passive stretching bout of the plantar flexor muscles.

The manuscript is on the whole well-written, the methods are technically sounding and the results adequately reported (although I suggest to use the exact P value and at least one measure of effect size to strengthen the results).

I still have some major questions the Authors should replied:

1) Please include and comment both in the Introduction and in the discussion a recent paper of Bisconti AV et al, J Phyisiol 2020 on this topic.

2) Was an a-prior sample size calculation performed for the two experimental set-up? if so, please include a paragraph.

3) Do the Authors feel that the Sit-and-Reach test would be sensitive enough to detect the changes in ROM after just an unilateral passives stretching bout of the plantar flexors? If not, the Authors should recognise that there is no parameter describing the stretch-induce change in ROM.

4) In the Discussion, please comment on the differences between your and Venturelli et al study.

Reviewer #2: The Authors studied the relationship between the relaxation period during stretching and the reduction of arterial stiffness, in healthy young men. In the first series of experiments they evaluated the changes in cardiac output, the shear rate in the posterior tibial artery and blood volume in the calf muscle during recovery (0-60 s) from a single bout of one-legged passive calf stretching. In the second series of experiments, they assessed the effects of different relaxation periods (5-, 10-, 20-, and 60-s) of passive one-legged intermittent calf stretching (30-s × 6 sets) on the femoral-ankle pulse wave velocity (faPWV) as an index of peripheral arterial stiffness. The results indicate that the stretched leg’s shear rate significantly increased from 0 to 10th s after stretching. The muscle blood volume in the stretched leg significantly decreased during stretching; then significantly increased during the recovery period after stretching. It was interesting that cardiac output remained unchanged during stretching and recovery. Moreover, the reduction in faPWV from the pre-stretching value in the stretched leg was significantly larger in the protocol with 10-s and 20-s relaxation periods than that in the non-stretched leg and in the 5-s and 60-s relaxation periods. The Authors suggest that the relaxation periods of intermittent static stretching causing a high transient increase in shear rate (via reperfusion after microvascular compression by the stretched calf muscles) are effective in reducing arterial stiffness. The study was well planned and the results are really interesting, demostrating that central hemodynamics are not really involved in the response to stretching in calf muscle arteries.

6. PLOS authors have the option to publish the peer review history of their article (what does this mean?). If published, this will include your full peer review and any attached files.

Reviewer #1: No

Reviewer #2: No

---

## [Author Response · Author response to Decision Letter 0]

27 Sep 2021

RESPONSE TO REFEREES (PONE-D-21-23457R1)

We wish to thank you for your constructive comments and suggestions. We have carefully considered all the comments and revised our manuscript accordingly. Our responses to each comment are as follows:

Reviewer #1: 

The manuscript is on the whole well-written, the methods are technically sounding and the results adequately reported (although I suggest to use the exact P value and at least one measure of effect size to strengthen the results).

Response: As you suggested, we have added the exact P values and effect size of the amount of change in faPWV in the Results section of the revised manuscript (Please see Page 12-13 and Page 11, Line 252-256).

(1) Please include and comment both in the Introduction and in the discussion a recent paper of Bisconti AV et al, J Phyisiol 2020 on this topic.

Response: Thank you for your suggestion. Accordingly, we have cited and discussed in the Introduction and Discussion sections of the paper of Bisconti AV et al, J Phyisiol 2020 (Please see Page 3, Line 49-50, and Page 17, Line 361-365).

(2) Was an a-prior sample size calculation performed for the two experimental set-up? if so, please include a paragraph.

Response: Thank you for your pointing it out. We have performed a-prior sample size calculation in this study. Therefore, we added these sentences in the Methods section (Please see Page 8, Line 187-189 and Page 11, Line 252-256).

(3) Do the Authors feel that the Sit-and-Reach test would be sensitive enough to detect the changes in ROM after just an unilateral passives stretching bout of the plantar flexors? If not, the Authors should recognise that there is no parameter describing the stretch-induce change in ROM.

Response: Thank you for your comments. As you suggested, in this study, the value of the sit-and-reach test did not differ between before and after the stretching, therefore, it was not an accurate test to assess the effect of flexibility by calf static stretching. Therefore, we have added this study limitation and the sentence of ‘Further study needs to examine measuring the range of motion of the ankle joint after calf static stretching’ in the text (Please see Page 19, Line 399-402).

(4) In the Discussion, please comment on the differences between your and Venturelli et al study.

Response: Thank you for your constructive comments. Accordingly, we have discussed the differences in the Discussion section (Please see Page 17-18, Line 370-372).

Thank you so much for your careful review and all your useful comments. We believe that your all comments improved this manuscript. We hope that you are satisfied with this revision.

Reviewer #2: 

The Authors studied the relationship between the relaxation period during stretching and the reduction of arterial stiffness, in healthy young men. In the first series of experiments they evaluated the changes in cardiac output, the shear rate in the posterior tibial artery and blood volume in the calf muscle during recovery (0-60 s) from a single bout of one-legged passive calf stretching. In the second series of experiments, they assessed the effects of different relaxation periods (5-, 10-, 20-, and 60-s) of passive one-legged intermittent calf stretching (30-s × 6 sets) on the femoral-ankle pulse wave velocity (faPWV) as an index of peripheral arterial stiffness. The results indicate that the stretched leg’s shear rate significantly increased from 0 to 10th s after stretching. The muscle blood volume in the stretched leg significantly decreased during stretching; then significantly increased during the recovery period after stretching. It was interesting that cardiac output remained unchanged during stretching and recovery. Moreover, the reduction in faPWV from the pre-stretching value in the stretched leg was significantly larger in the protocol with 10-s and 20-s relaxation periods than that in the non-stretched leg and in the 5-s and 60-s relaxation periods. The Authors suggest that the relaxation periods of intermittent static stretching causing a high transient increase in shear rate (via reperfusion after microvascular compression by the stretched calf muscles) are effective in reducing arterial stiffness. The study was well planned and the results are really interesting, demostrating that central hemodynamics are not really involved in the response to stretching in calf muscle arteries.

Response: Thank you so much for your careful review and all your constructive comments. We hope that you are also satisfied with this revision.

---

## [Decision Letter · Decision Letter 1]

20 Oct 2021

Acute effects of the different relaxation periods during passive intermittent static stretching on arterial stiffness

PONE-D-21-23457R1

Dear Dr. Iemitsu,

We’re pleased to inform you that your manuscript has been judged scientifically suitable for publication and will be formally accepted for publication once it meets all outstanding technical requirements.

Kind regards,

Vincenzo Lionetti, M.D., PhD

Academic Editor

PLOS ONE

Additional Editor Comments (optional):

Reviewers' comments:

Reviewer's Responses to Questions

**Comments to the Author**

1. If the authors have adequately addressed your comments raised in a previous round of review and you feel that this manuscript is now acceptable for publication, you may indicate that here to bypass the “Comments to the Author” section, enter your conflict of interest statement in the “Confidential to Editor” section, and submit your "Accept" recommendation.

Reviewer #1: All comments have been addressed

Reviewer #2: All comments have been addressed

2. Is the manuscript technically sound, and do the data support the conclusions?

Reviewer #1: Yes

Reviewer #2: Yes

3. Has the statistical analysis been performed appropriately and rigorously? 

Reviewer #1: Yes

Reviewer #2: Yes

4. Have the authors made all data underlying the findings in their manuscript fully available?

Reviewer #1: Yes

Reviewer #2: Yes

5. Is the manuscript presented in an intelligible fashion and written in standard English?

Reviewer #1: Yes

Reviewer #2: Yes

6. Review Comments to the Author

Reviewer #1: The Authors replied satisfactorily to all my points. In my opinion no further revisions are required.

Reviewer #2: The paper has been improved and can be accepted for publication. The Authors prepared a new manuscript implemented and acceptable. The Authors explained their changes in the text and substantially improved their presentation of the manuscript.

7. PLOS authors have the option to publish the peer review history of their article (what does this mean?). If published, this will include your full peer review and any attached files.

Reviewer #1: **Yes: **Emiliano Cè

Reviewer #2: No

---

## [Editor Report · Acceptance letter]

2 Nov 2021

PONE-D-21-23457R1 

Acute effects of the different relaxation periods during passive intermittent static stretching on arterial stiffness 

Dear Dr. Iemitsu:

I'm pleased to inform you that your manuscript has been deemed suitable for publication in PLOS ONE. Congratulations! Your manuscript is now with our production department. 

Kind regards, 

on behalf of

Prof. Vincenzo Lionetti 

Academic Editor

PLOS ONE